# *JAZ1-3* and *MYC2-1* Synergistically Regulate the Transformation from Completely Mixed Flower Buds to Female Flower Buds in *Castanea mollisima*

**DOI:** 10.3390/ijms23126452

**Published:** 2022-06-09

**Authors:** Hua Cheng, Sanxing Zha, Yanyan Luo, Li Li, Shiyan Wang, Shuai Wu, Shuiyuan Cheng, Linling Li

**Affiliations:** School of Modern Industry for Selenium Science and Engineering, Wuhan Polytechnic University, Wuhan 430048, China; chenghua1437@whpu.edu.cn (H.C.); sanxing_zha@126.com (S.Z.); yy_luo2022@163.com (Y.L.); lily7819@whpu.edu.cn (L.L.); 17839963906@163.com (S.W.); ws1490818609@163.com (S.W.); 12316@whpu.edu.cn (S.C.)

**Keywords:** *C. mollissima* BL., flower bud differentiation, Gibberellin, Methyl jasmonate, *MYC2*, *JA-ILE*

## Abstract

Chestnut (*Castanea mollisima*) is an important woody food crop, but its yield has been low in cultivation, mainly due to the problems of fewer female flowers and more male flowers. Therefore, regulating the transition of chestnut flowers and effectively balancing the proportion of male and female to improve the yield are key factor to be solved in production. In this study, the chestnut floral buds in pre- and post-winter were used as materials. The data of metabolites, hormones, and gene expression during flower bud differentiation of chestnut were analyzed by transcriptomics and metabolomics to preliminarily reveal the possible reason of male and female flower bud transformation in pre- and post-winter. The analysis of Differentially Expressed Genes (DEGs) showed that there were 6323 DEGs in the Complete mixed flower bud (CMF) group in pre- and post-winter, of which 3448 genes were up-regulated and 2875 genes were down-regulated. There were 8037 DEGs in the Incomplete mixed flower bud (IMF) in pre- and post-winter, of which 4546 genes were up-regulated and 3491 genes were down-regulated. A total of 726 genes from the two flower buds were enriched into 251 Kyoto Encyclopedia of Genes and Genomes (KEGG) pathways in post winter, of which plant hormone signal transduction accounted for 4.13%. The analysis results of differential metabolites showed that the differential metabolites of the two flower buds were mainly concentrated in the secondary metabolic synthesis pathway. The difference of hormone content showed that the content of Gibberellin 9 (GA9) and GA19 in CMF was higher than that in IMF in pre-winter, but the opposite in post-winter. Methyl jasmonate (MeJA) content was only very high in CMF in pre-winter, while Jasmonoyl-(l)-Isoleucine (JA-ILE) showed high content in CMF in post-winter. In post-winter, higher concentration of JA-ILE was positively correlated with the expression of Flowering Locus T (*CmFT*), and *CmFT* gene was significantly positively correlated with the expression levels of *MYC2-1*, *MYC2-2* and *LFY 3* (LEAFY 3). The higher concentration of JA-ILE was negatively correlated with the transcription level of *JAZ1-3*. In vitro experiments further verified that Jasmonate-Zim 1–3 (JAZ 1–3) combined with MYC2-1 inhibited the transcription of *CmFT* gene, while MYC2-1 alone promoted the expression of *FT*. The results suggested that a higher concentration of GA is conducive to breaking the dormancy of flower buds and promoting the development of male flower buds, while a lower concentration of GA and a higher concentration of JA-ILE are conducive to the differentiation and formation of female flower buds in post-winter, in which *JAZ1-3* and *MYC2-1* play a key role in the differentiation of female flower buds of chestnut.

## 1. Introduction

*Castanea mollissima* Blume, a chestnut plant of Fagaceae, is an excellent woody crop with high nutritional value and plays an important role in economy and ecology. However, the long-standing problems of excessive male flowers and insufficient female flowers (the ratio of male and female flowers is usually 1:2400–1:4000) in chestnut production have not been well solved, resulting in low per unit yield and poor benefits, which seriously restricts the healthy development of the chestnut industry [1,2]. Therefore, the exploration of flower bud differentiation mechanism is of great significance for regulating the proportion of male and female flowers, and provides valuable information for solving the problem of low yield of chestnut [3]. Flower development is mainly affected by genetic factors and environmental conditions. Therefore, the exploration of flower bud differentiation mechanism is of great significance for regulating the proportion of male and female flowers, and provides an important reference for solving the problem of low yield of chestnut [4]. Mechanistically, it is important to note that in monoecious species the development of male and female flower organs is genetically connected, which makes male and female flowers mutually exclusive. A high-level regulator differentially expressed in different parts of the same plant may be related to plant hormones and act as a switch between female and male development (master regulator of sex) [5]. Among them, phytohormones are essential for proper plant developmental regulation and have been extensively studied for their involvement in the flora transition [6,7]. Plant endogenous hormones can regulate plant growth and development by binding with specific protein receptors, and play a micro and efficient role in regulating flower bud differentiation of woody plants.

Among various phytohormones, GA plays a major role in affecting flowering, especially in floral sex determination in the model plant *Arabidopsis thaliana* [8]. GA signaling affects flowering not only through its interaction with known endogenous and environmental flowering genetic pathways, but also through its crosstalk with other phytohormones signaling pathway [9]. Auxin is one of the most well studied phytohormones. It is necessary for a variety of developmental activities, such as cell elongation and division, organ formation, root and bud development, and light and gravity response [10]. Auxin determines whether flower primordia are formed in the initial process of organs and the specificity of flower organs during organogenesis [11]. Jasmonic acid (JA) and its derivatives (named jasmonic acid) are fatty acid derived molecules that can regulate a variety of plant growth reactions, such as defense related reactions and developmental processes [12,13]. The key repressors of JA signaling are JAZ proteins [14]. Previous studies have shown that overexpression of the no degradable form JAZ leads to early flowering, indicating that JA signal cascade plays a negative role in flowering [15,16]. Abscisic acid (ABA) is generally considered as a stress-related phytohormones, and its role in flowering regulation is still controversial [17]. It has been reported that ABA has dual effects of flowering promotion and inhibition, and gibberellin and ABA play an antagonistic role in various developmental processes [18,19]. Ethylene is an important phytohormones, which is involved in regulating leaf senescence, fruit ripening, stress response, and flower transformation. Mutants with increased ethylene levels or constitutively activated ethylene signals delayed flowering under long and short-day conditions, supporting the role of ethylene as a flower inhibitor [20,21]. With the deepening of the research on phytohormones, it is found that it is not a certain endogenous hormone that directly determines the initiation of flowering differentiation in woody plants, but the dynamic balance of multiple hormones that regulates the normal progress of flower bud differentiation.

The latest progress in flowering regulation shows that DELLAs-mediated GA signal acts as a regulatory center to coordinate the cross-talking between various flowering genetic pathways, which can sense the environmental and developmental clues in leaf or stem apical meristem [9]. In particular, GA signals are strongly involved in the dynamic regulation of the expression of several flowering pathway genes, such as *FT,* suppressor of overexpression of constans 1 (*SOC1*) and *LFY* [22,23]. This allows the transformation of flowers to be strictly controlled by environmental signals and bioactive substances usually located in active proliferative tissues. Among the various molecules involved in GA pathway, DELLAs protein plays a major role in mediating the interaction between GA and key regulators in other flowering genetic pathways or phytohormones signaling pathways. The multiple mechanisms of DELLAs mediated transcriptional regulation allow flexible signal transduction between GA and other pathways [24]. It is worth noting that among various interacting protein molecules, DELLAs preferentially interacts with bHLH transcription factors, such as Brassinazole Resistant 1 (BZR1) in Brassinolide (BR) pathway, MYC2 and MYC3 in JA pathway and Phytochrome Interacting Factors (PIFS) in thermal sensing pathway [25,26]. In general, the GA signals involved in flower transformation should involve many intertwined feedback loops, which are related to various flowering cues and phytohormones signaling pathways in time or space [27]. Further clarifying the molecular interactions between known key participants and identifying new components in GA signals will expand our understanding of the molecular mechanism of flowering regulatory network.

Chestnut is a dioecious and monoecious fruit tree. According to the different growth and fruiting characteristics, the flower buds of chestnut can be divided into CMF, IMF, leaf buds, and auxiliary buds. Leaf buds and accessory buds belong to vegetative buds [28]. Both CMF and IMF belong to flower buds. The mixed buds of chestnut are variable and can be transformed under certain conditions. The inflorescence primordium differentiation stage pre-winter and post-winter has a great influence on the sex differentiation of chestnut flower buds, which is the key period for reasonable control. At present, there are few reports on the changes of phytohormones content and gene expression model related to flower bud differentiation in the key period of chestnut flower sex conversion. In this study, the mixed flower bud pre-winter and post-winter of female inflorescence primordia of chestnut were selected as materials. Through transcriptome sequencing and metabolome analysis, the key DEGs and differential metabolites involved in regulating the differentiation of female inflorescence primordia of chestnut were screened, and the five kinds of phytohormones (Auxin, Cytokinin, ABA, JA, and GA) were quantitatively detected to analyze the content changes of phytohormones in the process of female inflorescence primordium differentiation in chestnut mixed flower buds, in order to reveal the molecular mechanism of regulating the proportion of hermaphrodite inflorescences and the regulation mechanism of endogenous metabolites, especially endogenous hormones.

## 2. Results

### 2.1. Characteristics of Flower Buds of Chestnut Pre and Post Winter

The buds of chestnut can be divided into three types according to their properties: mixed flower buds, leaf buds, and auxiliary buds (Figure 1). In terms of bud size and morphology, the mixed flower bud is the largest, the leaf bud is the second, the auxiliary bud is the smallest, and the bud is covered with scales.

Mixed flower buds can be divided into CMF and IMF. CMF are attached to the top and lower nodes of branches (Figure 1(A1,A2)). The buds are hypertrophy and plump, blunt and round, less hairy, and the outer scales are large, which can wrap the whole bud, and develop into fruiting branches after germination. The IMF is inserted on the lower part of the CMF or on the branches with weak growth (Figure 1(B1,B2)). The bud is slightly smaller than the CMF, and the male flower branch is formed after germination. Nodes bearing CMF and IMF, without leaf buds. Therefore, after the inflorescence falls off, it forms a blind node and cannot branch.

Leaf buds, growing at the top and middle and lower parts of vigorous branches. The trees entering the fruiting stage are mostly planted in the middle and lower parts of various branches. The buds are smaller than IMF, nearly blunt triangular, and the two outer scales are smaller, which completely cover the two inner scales. After germination, various developmental branches are formed.

The accessory bud grows at the base of all kinds of branches at the shortened node. The bud is very small. It generally does not germinate but is in a dormant state and has a long service life. When the branches are broken or trimmed, it germinates only long branches.

### 2.2. Basic Data of Transcriptome Sequencing of Two Flower Buds

After filtering the original data, the high-quality clean reads of each sample reached more than 98%. In the high-quality clean reads, the percentage of Q20 base is more than 96%, Q30 base is more than 90%, and the content of GC is higher than 44%. These reference data show that the sequencing results are reliable and can be analyzed in the next step (Appendix A). The chestnut genome is used as the reference sequence for comparison [29], and the comparison results are shown in Appendix A.

The reads obtained from the feature count were compared with NCBI’s Non-redundant protein database (NR), Swissprot protein database, Clusters of orthologous groups for eukaryotic complete genomes (KOG) database, and KEGG protein database respectively for functional annotation and Gene Ontology (GO) enrichment analysis (Table 1).

### 2.3. KOG Functional Annotation of DEGs

The clean reads of the two mixed flower buds annotated to the KOG database in pre-winter. The results show that, the clear reads annotated as general function prediction are the most among the 13 KOG classifications, accounting for 47.27%. The second is signal transduction mechanism, accounting for 16.36%. The third is post-translational modification, protein turnover, and molecular chaperone, accounting for 9.09% (Appendix A). The clean reads of the two mixed flower bud post-winter were annotated into the KOG database and compared to 22 functional categories. Among them, the clear reads annotated with general function prediction are the most, accounting for 32.82%, the second is the signal transduction mechanism, with 123 clean reads, accounting for 23.61%, the third is post-translational modification, protein turnover, and molecular chaperone, accounting for 8.06% (Appendix A). The clean reads of CMF were annotated into the KOG database and classified into 24 functional categories. Among them, the clean reads with general functions are the most, accounting for 40.76%, the second is the signal transduction mechanism, with a total of 157 clean reads, accounting for 16.49%, the third is post-translational modification, protein turnover and molecular chaperone, accounting for 7.04% (Appendix A). The clean reads of IMF are annotated into the KOG database and classified into 24 functional categories. Among them, 36.88% of clean reads are annotated to general functions, the second is the signal transduction mechanism, with 152 clean reads, accounting for 12.95%, The third is post-translational modification, protein turnover and molecular chaperones, which have 116 clean reads, accounting for 9.88% (Appendix A).

### 2.4. GO Enrichment Analysis of DEGs in Two Mixed Flower Buds

GO functional cluster analysis was performed on the DEGs (differentially expressed genes) in the two mixed flower buds in pre-winter. The DEGs were mainly annotated into 31 functional categories of molecular function, cell component and biological process. The top five functions of each category are shown in Appendix A. The number of clean reads classified into molecular functions accounted for 16.20%, cell components accounted for 37.80%, and biological processes accounted for 46%. The GO functional classification results of the DEGs of two mixed flower buds of chestnut in post-winter showed that the number of clean reads classified into molecular functions accounted for 17.38%, cell components accounted for 40.03%, and biological processes accounted for 42.59% (Appendix A).

The GO functional classification of DEGs in the CMF of chestnut showed that the number of clean reads classified into molecular function accounted for 17.56%, the cellular component accounted for 41.85%, and the biological process accounted for 42.76% (Appendix A). The GO functional classification of DEGs in the transcriptome of IMF showed that the number of clean reads classified into molecular function accounted for 17.15%, the cellular component accounted for 40.28%, and the biological process accounted for 42.57% (Appendix A). Negative and positive regulation, stimulus response and signal transduction in biological processes, the enzyme regulatory factors, nucleotide binding transcription factors, hormone receptors, protein binding factors, translation regulatory factors, transcription factors, and other functional predictions related to temperature and light provide basic data for the further study of mixed flower bud differentiation in chestnut.

### 2.5. Analysis of KEGG Pathway of DEGs

In pre-winter, 162 differential clean reads were identified in the transcriptome of two type flower buds (A2 vs. B2), which were enriched in 67 metabolic pathways. Among them, phytohormones signal transduction and biosynthesis of amino acids accounted for the highest proportion, both of which were 6.17%, and there were 10 clean reads (Figure 2a).

In post-winter, 1696 differential clean reads were identified in the transcriptome of two type flower buds (A1 vs. B1), which were enriched in 109 metabolic pathways. Ribosome pathway accounted for the highest proportion, 5.19%, including 88 clean reads; phytohormones signal transduction took the second place, accounting for 3.77%, including 64 clean reads (Figure 2b).

A total of 2655 differential clean reads were identified in the CMF transcriptome in pre- and post-winter (A1 vs. A2), which were enriched in 108 metabolic pathways. It mainly includes carbon metabolism, ribosome, biosynthesis of amino acids, protein processing in endoplasmic reticulum, plant-pathogen interaction, and phytohormones signal transduction, accounting for 4.33%, 4.07%, 3.95%, 2.98%, 2.94%, and 2.79%, respectively (Figure 2c).

A total of 2892 differential clean reads were identified in the transcriptome of IMF in pre- and post-winter (B1 vs. B2), which were enriched in 108 metabolic pathways. These reads were mainly enriched in ribosome, carbon metabolism, biosynthesis of amino acids, phytohormones signal transduction, and plant–pathogen interaction, accounting for 4.84%, 3.98%, 3.67%, 3.11%, and 2.63% respectively (Figure 2d).

The above results showed that the Phytohormones signal transduction pathway had the largest number of DEGs between the two flower buds in pre- and post-winter, and a large number of DEGs were also enriched in the Phytohormones signal transduction pathway in two mix floral buds in pre- and post-winter.

There were 267 DEGs in CMF group and IMF group in pre-winter, of which 192 were up-regulated and 75 were down-regulated (Figure 3a). There were 3380 DEGs between CMF and IMF in post-winter, of which 1800 were up-regulated and 1580 were down-regulated (Figure 3b). There were 6323 DEGs in the CMF between pre- and post-winter, of which 3448 genes were up-regulated and 2875 genes were down-regulated (Figure 3c). There were 8037 DEGs in the IMF between pre- and post-winter, of which 4546 genes were up-regulated and 3491 genes were down-regulated (Figure 3d).

### 2.6. Confirmation of Expression Using qRT-PCR

To confirm the gene expression patterns identified by the transcriptome analysis, qRT-PCR assays were performed with the four sample (A1, A2, B1, and B2). Fourteen flowering related unigenes, including four hormone receptor related genes, five flowering pathway related genes, and five transcript factor related flowering regulate genes were selected to validate the RNA-seq data. The primer design sequence of candidate genes was listed in Appendix A. As expected, there was a good correlation between the qRT-PCR and RNA-seq data of the DEGs in the four libraries (Figure 4). These results suggested that transcriptome sequencing data could be used to further study the hormone regulated flowering pathway.

### 2.7. Differential Metabolite Screening

A large number of differential metabolites were obtained from the metabolome library constructed from the two types of mixed flower buds (Table 2, Appendix A, A2 vs. B2). There were 59 significantly different metabolites in the two flower buds in pre-winter (Figure 5a, Appendix A, A2 vs. B2). Among them, anthocyanins, indoles and derivatives, steroids, caffeic acid of phenylpropanoids, scopolamine, scopolamine, ferulic caffeic spermidine of phenolic amines, and other metabolites decreased significantly, while lipids, alcohols, nucleotides and derivatives, isoflavones, alkaloids, and other metabolites increased.

There were 90 significantly different metabolites between CMF and IMF in post-winter (Figure 5b, Appendix A, A1 vs. B1). Among them, polyphenols, flavonols, flavonoids, and terpenoids metabolites were significantly down-regulated, and amino acids and derivatives, alcohols, phenolic amines, nucleotides and derivatives, isoflavones, and organic acids were significantly up regulated.

There were 120 significantly different metabolites in CMF in pre- and post-winter (Figure 5c, Appendix A, A1 vs. A2), 16 metabolites such as polyphenols, sterides, β-caryophyllene and pelargonin were significantly down-regulated. Seventy-four metabolites such as alcohols, flavonols, vitamins and derivatives, and indole and derivatives were significantly up-regulated.

A total of 171 significantly different metabolites were found in IMF in pre- and post-winter (Figure 5d, Appendix A, B1 vs. B2), 34 metabolites such as flavonoids, quinones, vitamins and derivatives were significantly down-regulated, and 137 metabolites such as 2-aminoadipic acid, xanthine, anthocyanin, flavonol, and indole and derivatives were significantly up regulated.

### 2.8. Combined Analysis of Transcriptome and Metabolome

In order to compare the differences between the two mixed flower buds in the process of differentiation, we combined the transcriptome and metabolome of CMF and IMF in pre- and post-winter. The differential metabolites of CMF in pre- and post-winter are mainly concentrated in the metabolic pathway, and the differential genes are mainly concentrated in the pyrimidine metabolism (Figure 6a). The differential metabolites of IMF in pre- and post-winter period are mainly enriched in the metabolic pathway, and the differential genes are mainly enriched in the RAS signal pathway (Figure 6b).

In order to further display the correlation information between differential metabolites and differential genes, the analysis results use nine quadrants to reflect the correlation between genes and metabolites (Figure 7). There were 140 metabolites in fully mixed flower buds at two stages, which were positively correlated with their corresponding differential genes; when 701 metabolites were up-regulated, their corresponding genes were down-regulated or unchanged, showing a negative correlation; when 635 metabolites were down-regulated or unchanged, their corresponding genes were up-regulated, which was also negatively correlated (Figure 7a). A total of 183 differential metabolites were positively correlated with the corresponding differential genes in the two stages of incomplete mixed flower buds; 703 metabolites were down-regulated, and genes were up-regulated or unchanged; and 589 metabolites were down-regulated or changed, and genes were up-regulated (Figure 7b).

Thirty-four metabolites were obtained by comparing the CMF metabolome with the IMF metabolome in pre- and post-winter. Comparing the CMF transcriptome with the IMF transcriptome in pre- and post-winter, 686 genes were obtained by eliminating the same differential genes. The correlation analysis was carried out between these metabolites and genes. The correlation network diagram was established based on metabolites and genes as metabolism. They jointly participated in the Phytohormones signal transmission signal pathway (Figure 7). In this pathway, metabolites and genes were up-regulated, which was a positive correlation.

### 2.9. Analysis of Differential Hormone Content

The contents of six hormones in different samples were test and analyzed according to the metabolic pathway of Phytohormones. Taking the standard concentration (ng/mL) as the abscissa and the peak area as the ordinate, the standard curves of different hormones were drawn, and the linear regression equation was obtained (Appendix A). Through the quantitative analysis of various hormones, we found that there were significant differences in the contents of various hormones in the four samples (Figure 8, Appendix A).

In this experiment, the content changes of four auxin were detected, Indole-3-acetic acid (IAA), Methyl indole-3-acetate (ME-IAA), 3-Indolebutyric acid (IBA), and Indole-3-carboxaldehyde (ICA) (Figure 8a), IBA was not detected in the four groups of samples. Although the other three types of auxin were detected, there was no significant difference in the content of the four groups of samples, which showed that auxin did not play a key role in the process of chestnut flower bud differentiation, and it may interact with other hormones to play a role in flower inhibition and flower formation.

The detection results showed that the ABA content of CMF in pre-winter was much higher than that in post-winter (Figure 8b). This result indicated that the high content of ABA is helpful for the flower bud to enter the dormant state. After dormancy, the content of ABA decreases and the primordium differentiation of female inflorescence begins. It is speculated that the low content of ABA is helpful for the differentiation of female inflorescence.

The detection results of salicylic acid (SA) showed that the content of SA in IMF was higher than that in CMF in these two periods (Figure 8c). From the distribution of hormone content, SA was helpful to the formation of male flowers in the process of flower bud differentiation of chestnut.

Four cytokinin hormones, N6-Isopentenyladenine (IP), trans-Zeatin (tZ), cis-Zeatin (cZ), and dihydro-zeatin (dZ), were detected in four flower bud samples (Figure 8d). The results showed that the IP content of the two types of flower buds in pre-winter was significantly higher than that in post-winter, indicating that IP can promote the vegetative growth of flower buds. In post-winter, the IP content in CMF was higher than that in IMF, indicating that IP may promote the formation of female flowers. The content of tZ in the two types of flower buds in post-winter is much higher than that in pre-winter, indicating that tZ can promote flower bud differentiation, and the content in CMF is higher than that in IMF, indicating that high tZ content may be more conducive to the formation of female flowers. The content of cZ in IMF is higher than that in CMF in pre- and post-winter, indicating that cZ may be more conducive to the differentiation of male flowers. There was no significant difference in dZ content among the four groups.

The contents of JA, MeJA, Dihydrojasmonic acid (H2JA) and JA-ILE were determined and analysis in four samples (Figure 8e). The content of JA in pre-winter is higher than that in post-winter, which shows that JA is conducive to the vegetative growth of flower buds and accumulates certain nutrients for the later reproductive growth, which is related to JA can be conducive to the accumulation of sucrose in cells, promote the thickening of microtubules and microfilaments, and promote cell expansion. In pre-winter, the content of MeJA was very high in the CMF, but low in other samples. JA-ILE showed a high content in CMF in post-winter, suggesting that JA-ILE may be conducive to the transition of mixed flower buds into female flowers. There was no significant difference in the content of H2JA among the four groups.

A total of ten GA(Gibberellin) standard samples were detected, including GA1, GA3, GA4, GA7, GA9, GA15, GA19, GA20, GA24, and GA53. Among them, GA1, GA3, GA4, GA15, and GA24 were not detected in four samples (Figure 8f, Appendix A). GA7 was only detected in the two flower bud samples in post-winter, and the difference between the two samples was not significant, the results indicate that GA7 may promote reproductive growth in post-winter. The content of GA19 in flower buds in pre-winter is much higher than that in post-winter, and in CMF is higher than that in IMF, indicating that GA19 may play a key role in promoting the differentiation of female flowers. In pre-winter, the content of GA20 and GA9 in CMF is higher than that in IMF, but on the contrary in post-winter, indicating low content of GA53, GA20, and GA9 is conducive to the differentiation of female flowers in post-winter.

### 2.10. Combined Analysis of Flower Bud Differentiation Hormones and Related Genes

Combined with the changes of phytohormone content in CMF and IMF in pre- and post-winter, GA and JA were screened, and analyzed the relationship between hormone differences and corresponding expressed genes.

In the GA signal pathway (Figure 9a), CMHBY196485_GA20_1, CMHBY290603_GA20_8, CMHBY49384_GRP4 and other genes related to GA synthesis were screened from DEGs data. Cluster-49384.16068_P450 714C2, Cluster-49384.34987, Cluster-49384.25862_GAMYB genes related to GA degradation were screened from DEG database. The receptor protein genes related to GA regulation include CMHBY213795_GID1, CMHBY232612_GID2, CMHBY220381_Della1 and Cluster-49384_Della2 genes. The genes related to GA regulating downstream flowering include Cluster49384_constans-like1, Cluster49384_constans-like2, Cluster170750_constans-like3, Cluster49384_constans-like4, Cluster49384_constans-like5, Cluster49384_constans-like6, Cluster256120_constans-like7, Cluster49384_PIF4, Cluster232401_NF_Y_sB3, Cluster184700_NF_Y_SG, Cluster83615_NF_Y_SC2, novel.3632_MYC3-Like, Cluster49384_FLC. Genes related to flower bud differentiation and flowering include Cluster198372_FT, Cluster-49384.24265_SOC1. Genes related to JA pathway regulating flowering include CMHBY206099_JA-Z1, CMHBY213623_JA-Z2, CMHBY215636_JA-Z3, CMHBY213223_MYC2_1, CMHBY203600_MYC2_2.

Among the four flower bud samples, the difference of GA19 was the most significant (Figure 9b). In CMF, GA19 content reached 4.24 ng/g in pre-winter and rapidly decreased to 1.10 ng/g in post-winter. Other GA remained at a low level. In pre-winter, GA synthesis related gene *GA20_ 8* show a very high transcriptional level, but at a low transcriptional level in post-winter. In pre-winter, the content of GA in CMF is higher than that in IMF.

In IMF, the content of GA19 was 2.26 in pre-winter and decreased to 0.73 in post-winter. In different flower buds, the expression of *GA20_8* gene was consistent with the content of GA19. The rapid decrease in GA19 content led to the decrease in *GID1* and *GID2* transcription level, while *DELLA* transcription level increased in post-winter, which promoted the increase in *LFY*, *FLC,* and *MYC3* gene expression levels.

In post-winter, the total GA level of IMF was higher than that in CMF. The transcriptional levels of GA receptor genes *GID1*, *GID2*, and *Della* decreased in the two floral buds, while the transcriptional levels of genes such as *PIF4* and *GRP4* increased in the two floral buds. The transcriptional levels of downstream genes such as *FT*, *LFY1*, *TF1*, *SOC*, *CO*, and ELF only increased in CMF. The results suggest that the decrease in *GA20ox* transcription level and GA19 content play a major regulatory role in the development of female flowers, and play an important role in inhibiting the development of male flower buds and promoting the differentiation of female flower buds in post-winter. The relatively high levels of GA9, GA20, and GA53 in IMF may be related to the differentiation of male flower buds in post-winter.

Among the four flower bud samples, the content difference of JA-ILE was the largest (Figure 9c). The amount of JA-ILE in CMF was only 140.91 in pre-winter, but it quickly reached 448.85 in post-winter. The increase in JA-ILE content promoted the expression of *COI1*, *FT*, *LFY1,* and *TF1* and inhibited the expression of *JAZ1* and *JAZ2*. The decrease in *JAZ1* and *JAZ2* expression levels promoted the expression of *MYC2* and further promoted the increase in the transcription levels of *LFY1* and *FT* genes.

In post winter, the content of JA-ILE increased slightly in IMF, and the change was not significant (Figure 9c). In pre- and post-winter, JA and H2JA decreased in both flower buds. Low levels of JA-ILE, JA and H2JA resulted in a decrease in the expression levels of *COI1* and *JAZ1-3*, while the expression levels of downstream flowering related genes *FT*, *LFY1* and *TF1* decreased. The results suggest that a higher JA-ILE level regulates the decrease in *JAZ1* transcription level in CMF, resulting in *MYC2_ 1* and *MYC2_2* transcription increased. The higher level of *MYC2* promoted the expression of *FT*, *LFY1,* and *TF1*, thus promoting the expression of female flower related genes.

In order to further analyze the correlation of flowering related genes regulated by hormones, 50 flowering related genes and 8 plant hormone contents were selected to draw the co-expression networks (Figure 9d). Among them, *FT* gene and *MYC2_ 2*, *MYC2_ 1* and *LEAF3* have a significant positive correlation, and the correlation coefficients are 0.9998, 0.881553, and 0.85715, respectively. Among the hormone regulated flowering genes, *FT* was also highly correlated with JA-ILE, and the correlation coefficient was 0.97832. In post-winter, FT was negatively correlated with GA53 and GA20, and the correlation coefficients were −0.9402 and −0.86611, respectively. In the Methyl Jasmonate response pathway, the expression level of *JAZ1-3* gene was significantly negatively correlated with the plant hormone JA-ILE, and the correlation coefficient was −0.81116. It was significantly negatively correlated with the expression of *FT*, *MYC2-2,* and *LFY1*, and the correlation coefficients were −0.91168, −0.9176, and −0.99286, respectively.

### 2.11. JAZ1-3 and MYC2-1 Synergistically Regulate the Expression of FT

To examine the ability of *JAZ1-3* and *MYC2-1* to transactivate the promoters of ATC biosynthetic genes, transient dual-luciferase assays in tobacco leaves were carried out using the constructs shown in Figure 10a. Nucleotide logo of the predicted *MYC2-1* binding site in *FT* (Figure 10b) and *LFY1*(Figure 10c). When *MYC2-1* transfected alone, the activities of two segmented promoters of the *FT* genes were significantly increased by 37.6 and 31.37 folds to control. In addition, weaker activation of *P_CmFT_* was detected when tobacco leaves were simultaneously cotransformed with *CmJAZ1-3* and *CmMYC2-1*. When transformed with *JAZ1-3* alone, the transcriptional activation of *FT* gene was also weak, indicating that the *CmJAZ1-3* will inhibit the transcriptional activity of *MYC2-1* to *P_CmFT_*. When *MYC2-1* was transformed alone, p_Cmleafy2-674_ was significantly increased by 15.73 times compared with the control, while p_Cmleafy3_ was not significantly increased (Figure 10d).

In order to examine whether *JAZ1-3* homologs are able to interact with *MYC2-1*, we performed yeast two-hybrid assays (Figure 10e,f). Chestnut *MYC2-1* was used as a prey, and various *JAZ1-3* homologs were cloned as the bait. Transformed yeast cells growing on SD/-Leu -Trp selection medium. The results indicated that CmJAZ1-3 was able to interact with CmMYC2-1 in yeast.

## 3. Discussion

### 3.1. Critical Period of Flower Bud Differentiation in Chestnut

The vast majority of terrestrial plants independently evolved different genders in gametophyte (dioecious) or sporophyte (dioecious), but 43% of dioecious angiosperms only existed in 34 completely dioecious branches, indicating that their sex determination model evolved a long time ago [5]. The transition to flowering is a crucial step in the plant life cycle that is controlled by multiple endogenous and environmental cues, including hormones, sugars, temperature, and photoperiod [30]. The complex process of flower development is a dynamic reflection of the signal integration of external environment and internal factor [31]. In order to describe the characteristics of flower bud differentiation more accurately, flower bud differentiation can be subdivided into different periods. In *A. thaliana*, flower development was found to involve six pathways, the photoperiod, gibberellin, vernalization, autonomous, senescence, and ambient-temperature pathways. These pathways jointly regulate floral specific genes, leading to the physiological transformation of vegetative into flower meristem [32]. Flower bud differentiation of woody plants refers to the process of growth and transformation from vegetative tip to reproductive tip formed by woody plants every year, that is, the growth point on the branches of woody plants changes from vegetative buds to flower buds [33]. The floral development of chestnut shows differences from those of *A. thaliana*, which are characterized by monoecism (with unisexual flowers) and large differences between male and female flowers [2]. Chestnut is a monoecious and dioecious woody plant, which is more similar to the flower bud differentiation of *Poplar.* It is intriguing that the single-gene sex-determining system of *Poplars* may still exhibit some rudimentary manifestation of spatiotemporal control of sex expression, as exceptional female flowers on genetically male trees occur at the tip of the catkins, while exceptional male flowers on genetically female trees preferentially occur at its base [34].

The flower buds of chestnut can be divided into two types, CMF buds and IMF buds [35]. The CMF bud is inserted at the top and lower nodes of the branch. The bud is large and blunt, with a small amount of villi and large outer scales, which can wrap the whole bud. After germination, it is a fruiting branch, and there are male and female inflorescences (mixed inflorescences) on the new shoot at the same time; The ICM flower bud is inserted on the lower part of the CMF bud or on the branches with slightly weak growth. The bud is small and only germinates into a male inflorescence.

The flower bud differentiation of chestnut has particularity: first, the differentiation time of male and female flowers is inconsistent. The differentiation of male inflorescence will take place in the same year, and it will germinate and form inflorescence in March of the next year after dormancy in winter; Female inflorescence differentiation began only in late March to May of the next year, with the characteristics of extra bud differentiation and rare flowering. Second, both mixed flower buds of chestnut flower buds differentiate into male inflorescence primordia in pre-winter, while only the inflorescence primordia produced by CMF dormancy can differentiate into female flower cluster primordia in post-winter. It is often found that the emergence of two kinds of mixed flower buds of chestnut is often uncertain and shows a certain reversibility [36,37]. The pre-winter male inflorescence primordium of chestnut was formed from Sep to Dec of that year. This primordium differentiated and developed into male inflorescence after dormancy, entering the key dormancy period and inflorescence primordium formation period in post-winter [37]. Only the post-winter inflorescence primordium will have the potential to differentiate into female flower clusters in the future [35]. With the emergence of post-winter inflorescence primordium to female flower cluster primordium, it means the beginning of female flower differentiation. Therefore, it is determined that this period (from the completion of male inflorescence primordium differentiation to the beginning of female inflorescence primordium differentiation) is the key critical period for the differentiation of chestnut flower buds into female flowers [28].

### 3.2. Metabolites and Flower bud Differentiation

The metabolome analysis of two mixed flower buds obtained significantly different metabolites, which were mainly concentrated in the secondary metabolic pathway, amino acid biosynthesis, phytohormones biosynthesis, and other pathways. The data indicate that these metabolites play an important role in the flower bud differentiation of chestnut. The physiological differentiation of flower bud was earlier than the morphological differentiation. Its physiological differentiation is formed by the joint action of nutrients, phytohormones regulatory substances and genetic substances. During differentiation, various substances change from quantitative to qualitative, which provides a material basis for morphological differentiation [38].

In post-winter, the accumulation of higher contents of polyphenols, flavonols, vitamins and derivatives, lysine butyric acid, and other substances is conducive to the formation of female inflorescence, while lower contents of xanthine, lipid, alcohols and polyamines are conducive to the development of male inflorescence. Flower transition is related to hormones and polyamines, which has been proved in many plants. Exogenous polyamines can significantly promote apple flower bud differentiation, among which putrescine is considered to be the main polyamine for flower formation [39]. Spraying putrescine and spermidine could significantly increase the number of fruiting flowers and the proportion of male and female flowers [40]. Amino acids are the basis of protein synthesis. Free amino acids can provide nitrogen source for flower bud differentiation. In the four metabolic groups, we obtained a large number of differential metabolites of amino acids and derivatives. These metabolites are conducive to the differentiation of male inflorescence or the development of female inflorescence.

Some studies have shown that sugars, as signal molecules, regulate the flowering transition in rose and flowering induction in apple [41,42]. Carbohydrates are thought to play a crucial role in the regulation of flowering, and trehalose-6-phosphate (T6P) has been suggested to function as a proxy for carbohydrate status in plants [43]. Wang’s study indicated that in the process of flower bud differentiation, with the consumption of nitrogen, the concentration of nitrogen decreases gradually [44]. These high affinity nitrate transporters function in order to meet the need of nitrogen for flower organ development in later stage. In pre- and post-winter, KEGG analysis of differential genes showed that the main differential genes were enriched in carbon metabolism pathway, followed by biosynthesis of amino acid and phytohormones signaling pathway. This shows that carbon metabolism and hormone transduction pathway are very important for the development of two kinds of flower buds. While in post-winter, the differences between the two types of flower buds were mainly concentrated in the hormone metabolism pathway. In secondary metabolism, Hui’s study suggested that the male floral differentiation was associated with flavonoid biosynthesis process, but female floral transition was significantly involved in the phytohormone signal pathway [45]. In chestnut, less sugars and isoflavones were more conducive to the differentiation of female flowers in post-winter. While the accumulation of flavonoids, vitamins and derivatives, indole and derivatives were conducive to the formation of male inflorescence in post-winter.

### 3.3. GA and JA Jointly Regulate the Floral Transition in Chestnut

Monoecy is characterized by local signals controlling a high-level regulator determining the initiation of female or male developmental genetic networks (master regulator of sex) on different parts of the same plant. Hormones can control this process [5]. GA is an important hormone in plant growth and development and plays an important role in floral transition [2]. The internal environment stability of GAs is achieved by strictly regulating two “activating enzymes”, including GA20ox and GA3ox, which catalyze the last step of GA biosynthesis, and GA2ox, which promotes GA transformation [9]. Genetic and biochemical analysis of GA-signaling genes has revealed that posttranslational regulation of DELLA protein accumulation is a key control point in GA response. Active GA can be sensed and bound by the receptor GID1. When the concentration is high, GA enters the C-Port bag structure of GID1, the conformation of GID1 protein changes, and the N-terminal extension structure will cover the pocket structure to form a hydrophobic surface. This hydrophobic surface will promote the formation of GID1/GA/DELLA complex, thus relieving the inhibition of DELLA on key downstream regulators. Subsequently, this GID1-GA-DELLA complex stimulates binding of the E3 ubiquitin ligase SLY1 to DELLAs, leading to degradation of DELLAs by the 26S proteasome [46,47]. *FLC*, encoding MADS-box protein, suppresses many genes involved in floral induction such as *FT* and *SOC1* and is positively and negatively regulated through control of transcription by many regulators in a dosage-dependent pattern [48]. MYC3 is a bHLH transcription factor, which directly represses *FT* in a DELLA-dependent manner [49]. The degradation of DELLA protein leads to the inhibition and inactivation of *FLC* or *MYC3*, resulting in the increase in flowering gene expression, such as *ELF*. In *A. thaliana*, *ELF* encodes a circadian clock-regulated circadian clock function and flowering time [50].

In the flower bud development of chestnut pre-winter, the male inflorescence develops firstly [1,28]. Higher GA content increased the protein levels of GID1 and GID2 and decreased the transcriptional level of DELLAs. DELLAs/MYC3/FLC complex can inhibit the related genes of downstream flower bud differentiation. The lower expression level of *DELLA* leads to the degradation of DELLAs/MYC3/FLC complex and the inactivation of flower bud differentiation repressor protein MYC3/FLC, which promotes the transcription levels of *SOC1*, *ELF* in pre-winter (Figure 11a). These results suggested that it may be related to the breaking of dormancy of flower buds and the early development of male flowers in pre-winter.

Environmental and developmental signals regulate GA’s biosynthesis and signal transduction. GA’s transport and concentration play an important role in plant flowering [46]. Under warm ambient temperatures, a lower GA level significantly delayed the flowering time [51]. The lower GA level leads to the rapid increase in *DELLA* expression level, which can interact with another *FT* activator *PIF4* through the DNA recognition domain of PIF4 and prevent it from binding to the promoter of the target gene, as *FT* and *LFY* [52]. Therefore, DELLA may also mediate the expression of *FT* by blocking the transcriptional activation of *FT* by PIF4 in the warm environment in post-winter. In the incomplete mixed flower buds of chestnut, the decrease in transcription levels of flowering regulatory genes *FT* and *LFY1* may be related to the decrease in GA level and the increase in expression levels of *CO*, *PIF4,* and *NF-ys* in post-winter (Figure 11b).

The versatile mechanisms of DELLA- mediated transcriptional regulation allow flexible signal transduction between GA and other pathways, as JA pathway. JA controls a variety of transcription programs that affect plant growth, development, and response to biotic and abiotic stresses [12]. The transcription of JA response gene is activated during the accumulation of JA-ILE, and the production of JA-ILE is strictly controlled by environmental and developmental clues [53]. Studies have shown that DELLAs protein directly or indirectly regulates MYC2 function and may participate in JA signal and mediate the expression of *FT* in leaves under long and short day [49]. In the JA signal regulation pathway, DELLAs usually competes with MYC2 to bind JAZ1. When the content of GA is at a higher level, a small amount of DELLAs is not enough to bind JAZ1. Therefore, JAZ1 binding with MYC2 forms a complex to inhibit the expression of its downstream genes in the form of repressors. When the content of GA decreases, a large number of DELLAs proteins bind to JAZ1, so that MYC2 can be released from JAZ1-MYC2 complex. MYC2 is activated and bound to the regulatory region of target gene to start the expression of downstream flowering gene (Figure 11c) [9,54]. The content of JA-ILE increases significantly and the GAs decreases in CMF in post-winter, which lead to the transcription level of *JAZ1* decreases and the expression level of *DELLAs* increases. The higher DELLAs protein binds to JAZ1 in CMF in post-winter. This competitive binding enables the JAZ1-MYC2 complex to release MYC2, and the higher transcriptional level of *MYC2* initiates the expression of genes related to female flower development, such as *TF1*, *LFY1,* and *FT.* In IMF buds, JA-ILE decreased significantly, and GAs level changed slightly, resulting in higher *JAZ1* expression and unable to fully release MYC2. Therefore, the expression of genes related to female flower development was still inhibited in post-winter.

Although the association analysis of differentially expressed genes and metabolites showed the correlation between hormones and molecular mechanisms of flower bud differentiation in dioecious plants, these predictions wait to be contrasted in more systematic ways with genome-wide selection scans results, genomic prediction estimates, and machine learning [55]. In addition, it is envisaged that in the future, GWAS and GP will be combined with gene editing technology to edit the related genes of DEGs traits, so as to improve the yield of woody plants, shorten the juvenile period of fruit trees, and change the sterility of trees, so as to achieve the goal of tree breeding [56,57].

## 4. Materials and Methods

### 4.1. Plant Materials

The test materials were taken from east chestnut garden in Luotian (31.021052, 115.595175), Hubei Province, China, Grafted seedlings of the same Chestnut Variety. The flower buds of chestnut in the early stage of winter dormancy were selected when the differentiation of male inflorescence primordia was completed and the female inflorescence primordia did not appear (17 December 2017). In post-winter, the flower buds of chestnut with complete differentiation of female inflorescence primordia were used as test samples for transcriptome and metabolome after critical period (15 March 2018).

The CMF (complete mixed flower buds, the second bud of the mother branch from top to bottom) and IMF (Incomplete mixed flower buds, the second bud of the mother branch from bottom to top) of chestnut trees with basically the same age were selected in pre- and post-winter (Figure 1). Ten mixed samples were taken for each bud, and each mixed sample was repeated three times. Three samples of CMF pre-winter were labeled A2-1, A2-2, and A2-3, respectively. Three samples of CMF post-winter were labeled A1-1, A1-2, and A1-3 respectively. IMF in pre-winter were labeled B2-1, B2-2, and B2-3, respectively. IMF in post-winter were labeled B1-1, B1-2, and B1-3 (Appendix A). Flower buds were put in an EP tube, wrapped with tin foil, immediately put into liquid nitrogen for frozen storage, then taken back to the laboratory and stored in a refrigerator at −80 ℃.

### 4.2. Chemicals and Reagents

HPLC-grade acetonitrile and methanol were purchased from Merck (Darmstadt, Germany). Milli Q water (Millipore, Bradford, USA) was used in all experiments. All of the standards were purchased from Olchemim Ltd. (Olomouc, Czech Republic) and Sigma (St. Louis, MO, USA). Acetic acid was obtained from Sinopharm Chemical Reagent (Shanghai, China). The stock solutions of standards were prepared at the concentration of 10 mg/mL in ACN. All stock solutions were stored at −20 °C. The stock solutions were diluted with ACN to working solutions before analysis.

### 4.3. RNA Extraction, Clustering, Sequencing, and Quality Control

Total RNA of four stages were extracted by the Trizol Kit (Promega, Beijing, China) according to the manufacturer’s instructions. The RNA was treated with DNase I (Takara to remove genomic DNA). The RNA integrity and quality were verified by RNase-free agarose gel and NanoDrop 2000 (IMPLEN, Westlake Village, CA, USA).

A total amount of 1 µg RNA per sample was used as input material for the RNA sample preparations. Sequencing libraries were generated using NEB Next^®^ Ultra TM RNA Library Prep Kit for Illumina^®^ (NEB, Ipswich, MA, USA) following manufacturer’s recommendations and index codes were added to attribute sequences to each sample. mRNA was purified from total RNA using poly-T oligo-attached magnetic beads. Fragmentation was carried out using divalent cations under elevated temperature in NEB Next First Strand Synthesis Reaction Buffer (5X). First strand cDNA was synthesized using random hexamer primer and M-MuLV Reverse Transcriptase (RNase H^−^). Second strand cDNA synthesis was subsequently performed using DNA Polymerase I and RNase H. Remaining overhangs were converted into blunt ends via exonuclease/polymerase activities. After adenylation of 3′ ends of DNA fragments, NEB Next Adaptors with hairpin loop structure were ligated to prepare for hybridization. In order to select cDNA fragments of preferentially 250~300 bp in length, the library fragments were purified with ^AM^Pure XP system (Beckman Coulter, Beverly, USA). Then 3 µL USER Enzyme (NEB, USA) was used with size-selected, adaptor-ligated cDNA at 37 °C for 15 min followed by 5 min at 95 °C before PCR. Then PCR was performed with Phusion High-Fidelity DNA polymerase, Universal PCR primers and Index (X) Primer. Finally, PCR products were purified, and library quality was assessed on the Agilent Bioanalyzer 2100 system.

The clustering of the index coded samples was performed on a cBot Cluster Generation System using TruSeq PE Cluster Kit v3-cBot-HS (Illumina, San Diego, CA, USA) according to the manufacturer’s instructions. After cluster generation, the library preparations were sequenced on an Illumina Novaseq platform and 150 bp paired-end reads were generated.

Raw data (raw reads) of fastq format were firstly processed through in-house perl scripts. In this step, clean data (clean reads) were obtained by removing reads containing adapter, reads containing ploy-N and lower quality reads from raw data. At the same time, Q20, Q30, and GC content the clean data were calculated. All the downstream analyses were based on the clean data with high quality.

### 4.4. De Novo Assembly, Functional Annotation and DEG Analysis

The mapped reads of each sample were assembled by String Tie (v1.3.3b) in a reference-based approach [58]. String Tie uses a novel network flow algorithm as well as an optional de novo assembly step to assemble and quantitate full-length transcripts representing multiple splice variants for each gene locus.

Feature Counts v1.5.0-p3 was used to count the reads numbers mapped to each gene. Then the FPKM of each gene was calculated based on the length of the gene and the reads count mapped to this gene. FPKM, expected number of fragments per kilobase of transcript sequence per million base pairs sequenced, considers the effect of sequencing depth and gene length for the reads count at the same time, and is currently the most commonly used method for estimating gene expression levels.

Differential expression analysis of two groups (two biological replicates per condition) was performed using the DESeq2 R package (1.16.1). DESeq2 provide statistical routines for determining differential expression in digital gene expression data using a model based on the negative binomial distribution. The resulting P-values were adjusted using the Benjamini and Hochberg’s approach for controlling the false discovery rate. Genes with an adjusted *p*-value < 0.05 found by DESeq2 were assigned as differentially expressed.

GO enrichment analysis of differentially expressed genes was implemented by the cluster Profiler R package, in which gene length bias was corrected. GO terms with corrected *p*-value less than 0.05 were considered significantly enriched by differential expressed genes. KEGG is a database resource for understanding high-level functions and utilities of the biological system, such as the cell, the organism and the ecosystem, from molecular-level information, especially large-scale molecular datasets generated by genome sequencing and other high-through put experimental technologies (http://www.genome.jp/kegg/ accessed on 5 September 2019). We used cluster Profiler R package to test the statistical enrichment of differential expression genes in KEGG pathways.

### 4.5. Real-Time PCR Validation

The qRT-PCR was performed using cDNA templates. Each cDNA of the mRNA was amplified by qPCR using 2× TSINGKE Master qPCR Mix (SYBR Green I) and Goldenstar™ RT6 cDNA Synthesis Mix (TSINGKE, Beijing, China).

Applied Biosystems, Foster City, CA USA) using the SYBR-Green (Takara, Dalian, China) method. The primers were designed using SnapGene (version 3.5, Appendix A). Each reaction was carried out with a volume of 20 μL, which contained 10 μL SYBR, 8.6 μL ddH_2_O, 1 μL diluted template (1 μL of the generated first-strand cDNA diluted by 9 μL ddH_2_O) and 0.2 μL of each of two gene specific primers. The following program was used conditions: 94 °C for 30 s (pre-denaturation) followed by 40 cycles at 94 °C for 30 s (denaturation), 60 °C for 20 s (primer annealing) and 72 °C for 43 s (extension and gathering the fluorescent signal). At the end, the melting curve analysis was executed for verifying the specificity of the primer with the following program: 95 °C/15 s, 60 °C/1 min, and 95 °C/15 s. The baseline and threshold cycles (Ct) were automatically determined by the own software of the system. Transcript levels were normalized against the average expression of the Actin gene.

### 4.6. Metabolome Sample Preparation and Extraction

The freeze-dried leaf was crushed using a mixer mill (mm 400, retsch) with a zirconia bead for 1.5 min at 30 hz. 100 mg powder was weighted and extracted overnight at 4 °C with 1.0 mL 70% aqueous methanol. Following centrifugation at 10, 000g for 10 min, the extracts were absorbed (CNWBOND Carbon-GCB SPE Cartridge, 250 mg, 3ml; ANPEL, Shanghai, China) and filtrated (SCAA-104, 0.22μm pore size; ANPEL, Shanghai, China) before LC-MS analysis.

### 4.7. HPLC Conditions

The sample extracts were analyzed using an LC-ESI-MS/MS system (HPLC, Shim-pack UFLC SHIMADZU CBM30A system, www.shimadzu.com.cn/ accessed on 11 February 2019; MS, Applied Biosystems 6500 Q TRAP, www.appliedbiosystems.com.cn/ accessed on 17 February 2019). The analytical conditions were as follows, HPLC: column, Waters ACQUITY UPLC HSS T3 C18 (1.8 µm, 2.1 mm × 100 mm); solvent system, water (0.04% acetic acid): acetonitrile (0.04% Acetic acid); gradient program,95:5 *v*/*v* at 0 min, 5:95 *v*/*v* at 11.0 min, 5:95 *v*/*v* at 12.0 min, 95:5 *v*/*v* at 12.1 min, 95:5 *v*/*v* at 15.0 min; flow rate, 0.40 mL/min; temperature, 40 °C; injection volume: 2 μL. The effluent was alternatively connected to an ESI-triple quadrupole-linear ion trap (Q TRAP)-MS.

### 4.8. ESI-Q TRAP-MS/MS

LIT and triple quadrupole (QQQ) scans were acquired on a triple quadrupole-linear ion trap mass spectrometer (Q TRAP), API 6500 Q TRAP LC/MS/MS System, equipped with an ESI Turbo Ion-Spray interface, operating in a positive ion mode and controlled by Analyst 1.6.3 software (AB Sciex, Framingham, MA, USA). The ESI source operation parameters were as follows: ion source, turbo spray; source temperature 500 °C; ion spray voltage (IS) 5500 V; ion source gas I (GSI), gas II (GSII), and curtain gas (CUR) were set at 55, 60, and 25.0 psi, respectively; the collision gas (CAD) was high. Instrument tuning and mass calibration were performed with 10 and 100 μmol/L polypropylene glycol solutions in QQQ and LIT modes, respectively. QQQ scans were acquired as MRM experiments with collision gas (nitrogen) set to 5 psi. DP and CE for individual MRM transitions were performed with further DP and CE optimization. A specific set of MRM transitions were monitored for each period according to the metabolites eluted within this period.

### 4.9. Yeast Two-Hybrid Assays

The coding sequences of CMHBY206099 was cloned into bait plasmid pGBKT7 vectors (Clontech, Terra Bella Ave, Mountain View, CA, USA), while CMHBY213223 was inserted into a pGADT7 vector (Clontech). The primers used are listed in Appendix A. The resulting pGBKT7-CmJAZs constructs were cotransformed together with pGADT7-CmMYC2-2 into Y2H Gold cells according to the Yeast transformation system 2 protocol. The transformed cells were subsequently incubated on tryptophan and leucine SD media and on Tryptophan, Leucine, Histidine, and Adenine SD media in the presence or absence of X-α-Gal. pGBKT7-53 and pGADT7-T served as positive controls, and pGBKT7-Lam and pGADT7-T served as negative controls [59].

### 4.10. Dual-Luciferase Assay of Transiently Transformed Tobacco Leaves

The full-length CDS of *JAZ1-3*-like gene and *MYC2-1*-like gene were isolated from the post-winter flowering bud of chestnut and cloned into the multiple cloning sites of the pGreenII 62-SK vectors. Segmented promoters of Cluster-198372.0 (*FT*-like gene) containing putative *MYC2* elements were inserted into the cloning site of pGreen II 0800-LUC. All constructs were transformed into *Agrobacterium* strain GV3101(p-Soup). Agrobacterium culture mixtures of TFs (*JAZ1-3L*, *MYC2-1L*) and promoters (10:1) were infiltrated into 4-week-old tobacco (*N. benthamiana*) leaves by using needleless syringes. The infiltration, transient expression analysis, and enzyme activity quantification of Firefly luciferase (LUC) and Renilla luciferase (REN) were conducted. Three days after infiltration, LUC and REN activities were analyzed using a Dual-Luciferase Reporter Assay System (Promega, WI, USA).

Finally, TF promoter interactions were measured as the ratio of LUC to REN in three independent experiments with at least three biological replicates for each assay. The primers used for the dual-luciferase assay are shown in Appendix A.

### 4.11. Statistical Analysis

Excel 2021 and SPSS (22.0) were used for experimental data processing. The heat map of expression analysis and hormone content analysis were processed by TB-tools software [60]. Duncan’s test was used to analyze significant differences (*p* ≤ 0.05). All of the experimental data had three biological replicates. Correlation Network was performed using the OmicStudio tools at https://www.omicstudio.cn/tool accessed on 21 November 2021. The positive correlation threshold is set to be greater than or equal to 0.5, the negative correlation threshold is set to be less than or equal to—0.5, and the *p*-value threshold is less than 0.5, R version 3.6.1, igraph1.2.6.

## 5. Conclusions

Studying the genetics of adaptation to new environments in ecologically and industrially important tree species is currently a major research line in the Ffields of plant science and genetic improvement for tolerance to abiotic stress [61]. In wood plants, the transformation of flowers is flexibly controlled by various environmental conditions and endogenous developmental clues. Plants constantly integrate information from the environment to adjust their growth, development, and defense capabilities in an adaptive way. Most of this phenotypic plasticity is regulated by the synergistic effect of a trace amount of phytohormones. The transformation of flower buds of woody plants is more complex. We selected the CMF and IMF of chestnut as the research object, and studied the influencing factors of flower bud differentiation in the two key periods pre- and post-winter through transcriptome, metabolome, and hormone analysis. The results showed that the higher content of GA in the two flower buds was conducive to breaking the dormancy of flower buds in pre-winter, promoting the expression of flowering related genes, such as *LEAFY*, *CO*, and *ELF*, and conducive to the early development of flower buds and the accumulation of nutrition. After overwintering, lower concentration of GA and higher concentration of JA-ILE in CMF promoted the degradation of *JAZ* and alleviated the inhibition of *MYC2-1* and *FT*, which was conducive to the differentiation and formation of female flower buds of chestnut. *FT* gene played a key role in the differentiation of female flower buds of chestnut. In IMF, relatively low concentration of JA-ILE and higher concentration of GA promoted the expression of *DELLAs* and inhibited the expression of *MYC3* and *FLC*, which was conducive to the continuous development and vegetative growth of male flowers. These research data are expected to provide theoretical guidance for chestnut production. It is possible to regulate the differentiation from CMF to female flower buds by exogenous MeJA and gibberellin, so as to improve the yield of chestnut.

## Figures and Tables

**Figure 1 ijms-23-06452-f001:**
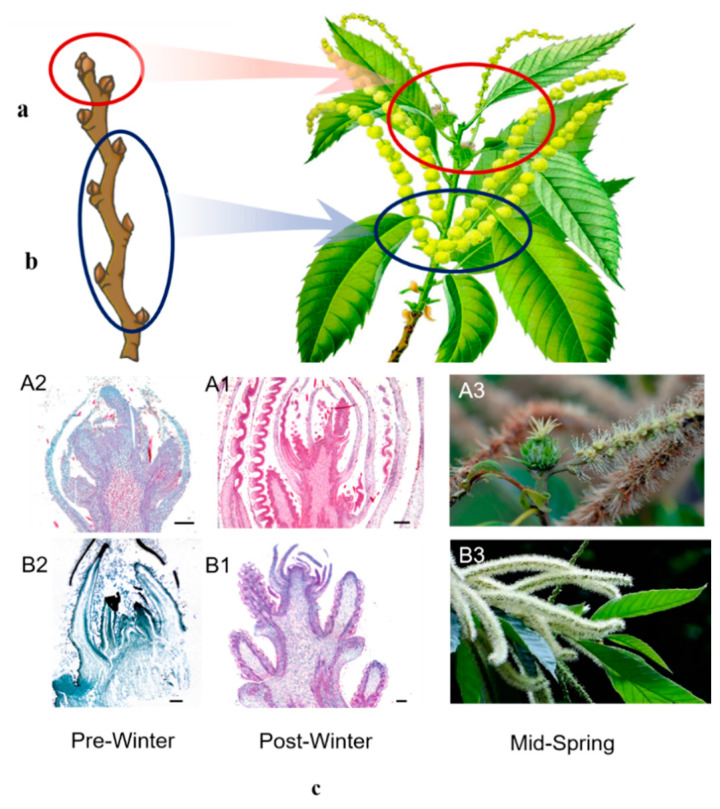
Slice diagram of chestnut flower bud in pre- and post-winter. Black bar represents 300 µm. (**a**) Vegetative branch. The buds are arranged on the left and right sides to express the size and coverage of buds and bud scales in different parts; (**b**) Fruiting branch. The left and right pairs of small buds at the bottom are auxiliary buds, 1–4 are branch small buds, 5–16 are incomplete mixed flower buds, 15–16 are mixed flower buds, and 18–19 are large buds above the male inflorescence; (**c**) Two different mixed flower buds of Chestnut. A1 is the CMF post-winter. A2 is the CMF in pre-winter, and A3 is the CMF in flowering. B1 is IMF in post-winter. B2 is the IMF in pre-winter, and B3 is the IMF in flowering.

**Figure 2 ijms-23-06452-f002:**
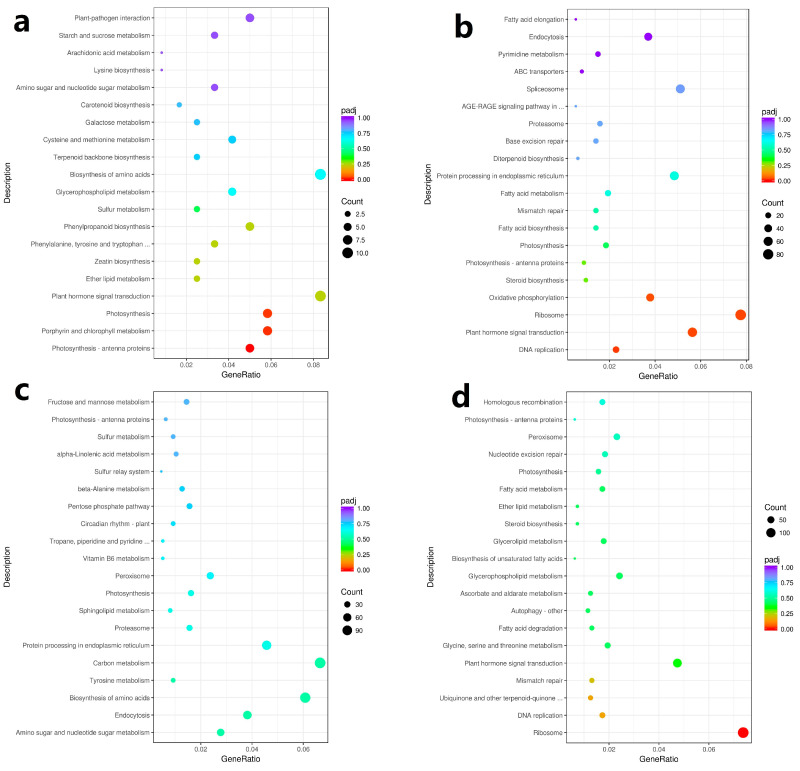
KEGG pathway analysis of mixed flower bud in pre- and post-winter chestnut. According to the results of DEGs, the KEGG pathway is enriched and the bubble diagram is drawn. The abscissa in the figure is the ratio of the number of differential genes annotated to the KEGG pathway to the total number of differential genes, the ordinate is the KEGG pathway, the size of the point represents the number of genes annotated to the KEGG pathway, and the color from red to purple represents the significance of enrichment. (**a**) Two kinds of chestnut mixed flower buds in pre-winter; (**b**) two mixed flower buds of chestnut in post-winter; (**c**) the CMF in pre- and post-winter; (**d**) the IMF in pre- and post-winter.

**Figure 3 ijms-23-06452-f003:**
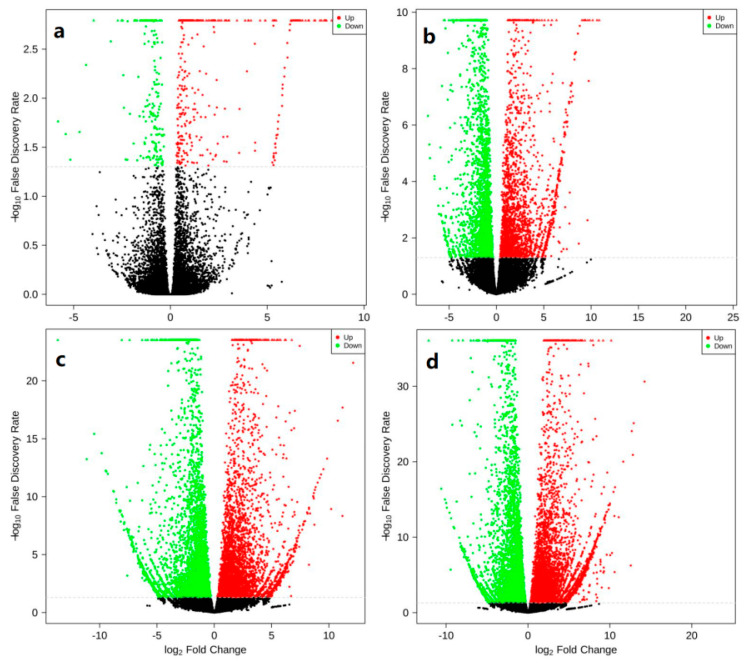
Volcanic map of mixed flower buds of chestnut in pre- and post-winter. The abscissa represents the change of gene expression multiple, and the ordinate represents the significance level of differential genes. Red is up-regulated, green is down-regulated, and black is non differentially expressed. (**a**) Mixed flower buds of two kinds of chestnut in pre-winter; (**b**) two mixed flower buds of chestnut in post-winter; (**c**) the CMF in pre-winter and post-winter; (**d**) the IMF in pre-winter and post-winter.

**Figure 4 ijms-23-06452-f004:**
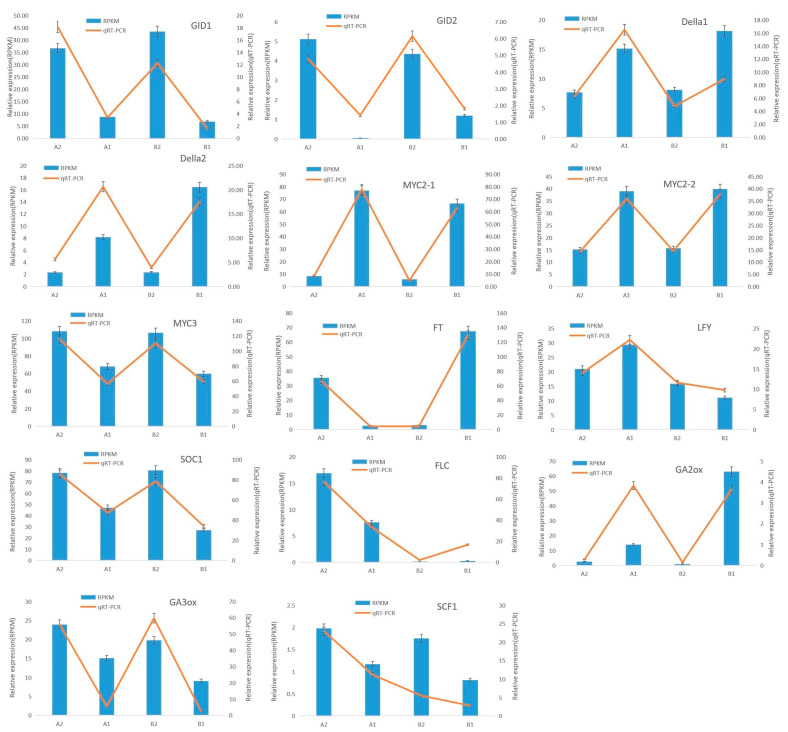
Comparative analysis of qRT-PCR and RNA-seq of candidate genes related to flower bud differentiation in chestnut. Actin was used as the control. The error bars represent the SD of three biological replicates. The numbers above the graphics correspond to values obtained with the correlation analysis of the gene expression ratios obtained from the RNA-seq data (column) and the qRT-PCR data (fold line). A1. The CMF in post-winter. B1. The IMF in post-winter. A2. The CMF in pre-winter. B2. The IMF in pre-winter.

**Figure 5 ijms-23-06452-f005:**
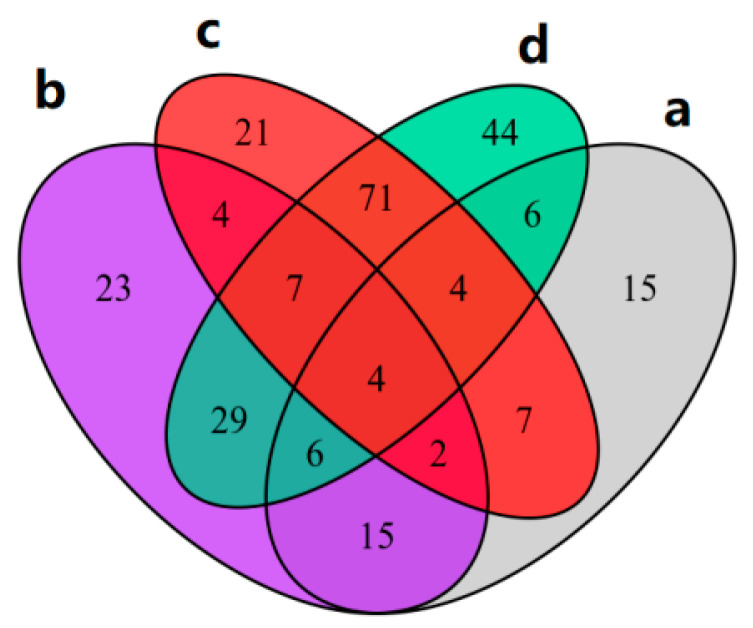
Venn diagram of different metabolites in the mix flower buds of each group. (**a**) Two mixed flower buds in pre-winter; (**b**) Two mixed flower buds in post-winter; (**c**) The CMF in pre- and post-winter; (**d**) The IMF in pre- and post-winter.

**Figure 6 ijms-23-06452-f006:**
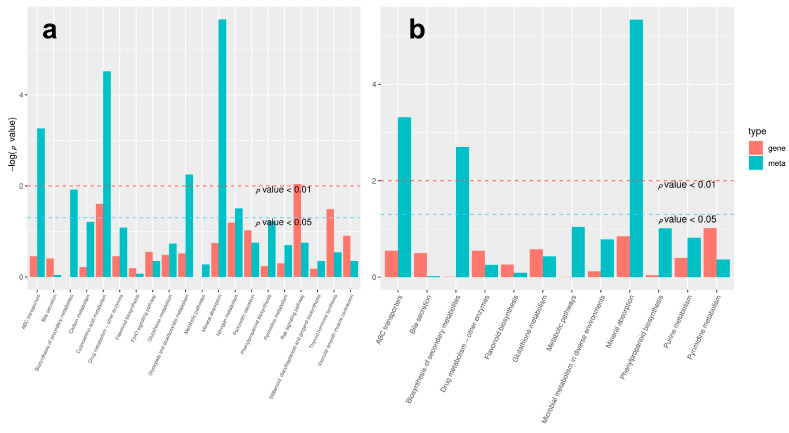
Enrichment map of differential metabolites and differential genes in mix flower buds of chestnut. (**a**) The CMF in pre- and post-winter; (**b**) The IMF in pre- and post-winter.

**Figure 7 ijms-23-06452-f007:**
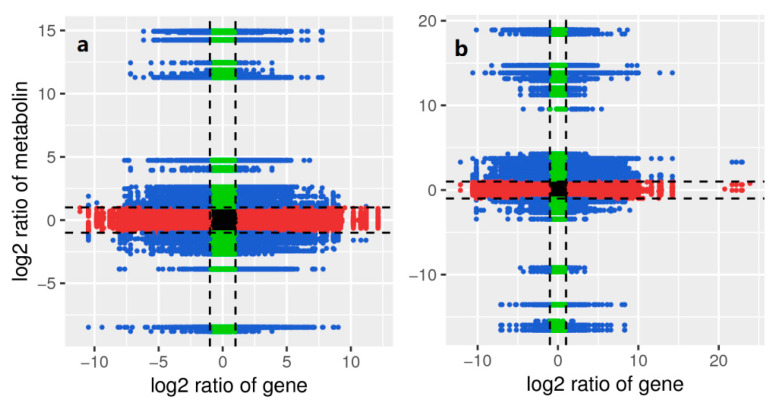
Correlation between different metabolites and genes in flower bud in pre- and post-winter. With the black dotted line as the boundary, it is divided into 1–9 quadrants from left to right and from top to bottom, Quadrants 3 and 7 represent positive correlation between genes and metabolites. The change of metabolites may be positively regulated by genes, and the differential expression pattern of genes and metabolites is consistent. Blue represents differential metabolites and expressed genes, green represents non differential metabolites, and red represents non differential expressed genes. (**a**) The CMF in pre- and post-winter; (**b**) the IMF in pre- and post-winter.

**Figure 8 ijms-23-06452-f008:**
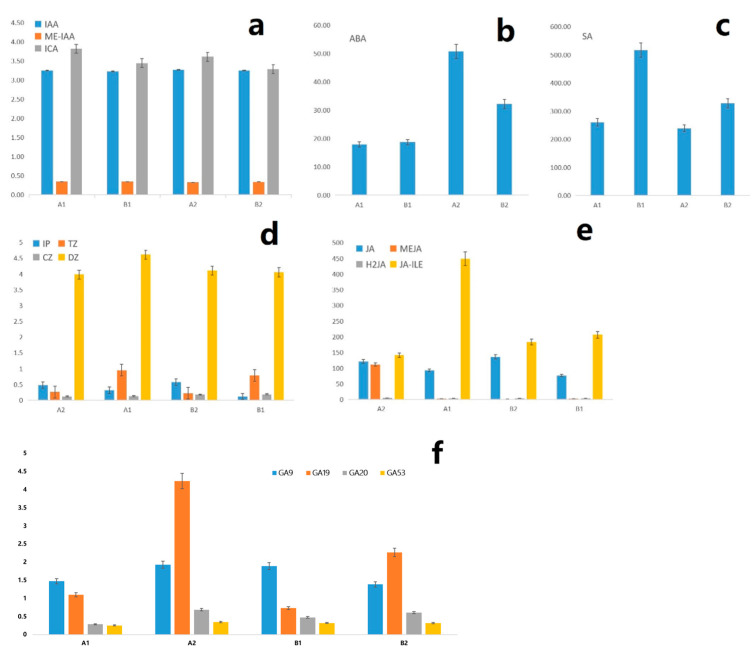
Difference of phytohormones content between the CMF and IMF of C. mollissima. A1. The CMF in post-winter; B1: The IMF in post-winter; A2: The CMF in pre-winter; B2: The IMF in pre-winter. (**a**) Changes of auxin content in different flower buds; (**b**) Change of abscisic acid content; (**c**) Change of salicylic acid content; (**d**) Change of cytokinin content; (**e**) Changes of methyl jasmonate content; (**f**) Change of gibberellin content.

**Figure 9 ijms-23-06452-f009:**
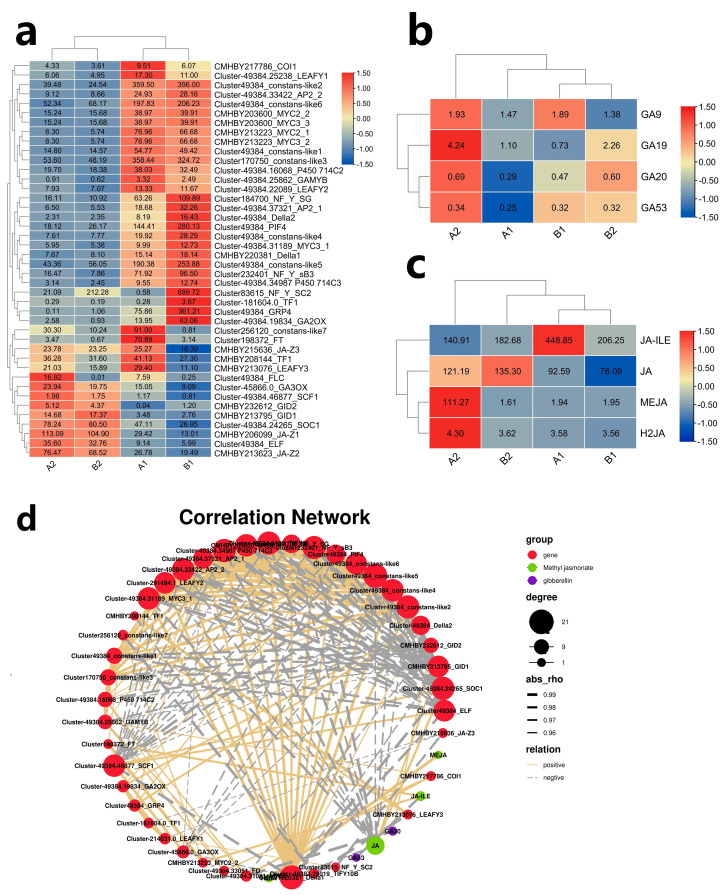
Changes of phytohormones content and expression of related genes in two different mixed flower buds of *C. mollissima*. A1. The CMF in post-winter; B1: The IMF in post-winter; A2: The CMF in pre-winter; B2: The IMF in pre-winter. (**a**) Analysis on the expression of flower bud differentiation related genes in complete mixed flower buds and incomplete mixed flower buds of chestnut at different stages; (**b**) The GAs content of two mixed flower buds in pre- and post-winter; (**c**) Difference analysis of MeJA content in different flower buds at different stages; (**d**) Regulating network between the selected genes and phytohormones compounds in chestnut flowering bud. The positive correlation threshold is set to be greater than or equal to 0.5, the negative correlation threshold is set to be less than or equal to −0.5, and the *p*-value threshold is less than 0.5, R version 3.6.1, igraph1.2.6.

**Figure 10 ijms-23-06452-f010:**
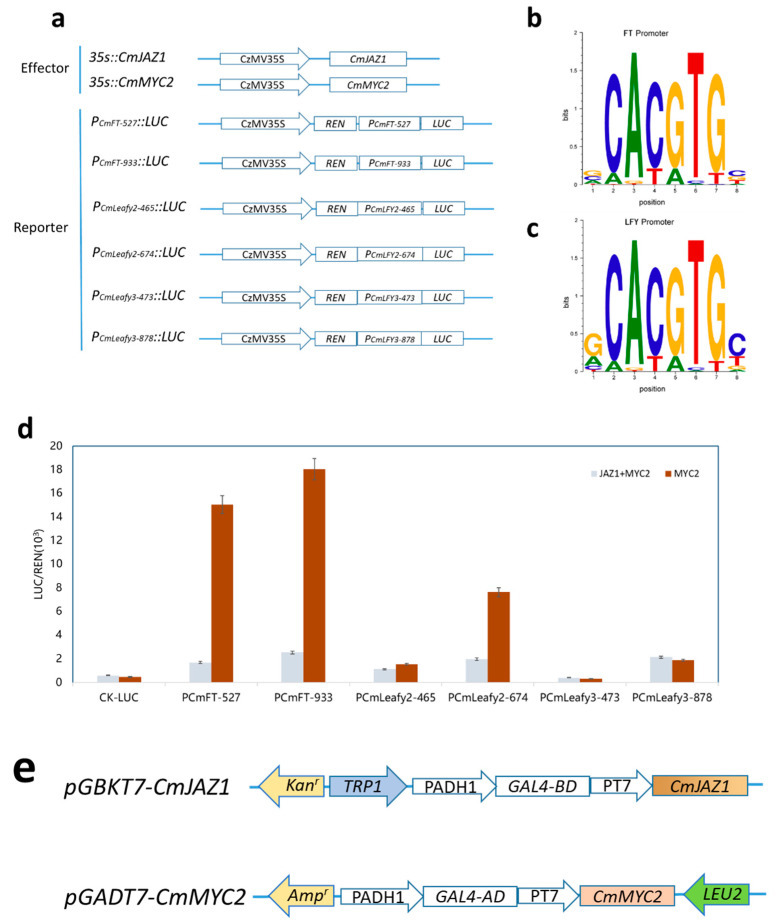
In vitro validation of interaction between CmMYC2 and CmJAZ1 in regulating *FT* and *LFY* gene expression. (**a**) Construct details for dual-luciferase assays; (**b**) Nucleotide logo of the predicted MYC2-1 binding site in *FT* genes; (**c**) Nucleotide logo of the predicted MYC2-1 binding site at the regulating area of *LFY*; (**d**) The in vivo associations of TFs (CmMYC2 and CmJAZ1) and promoters were obtained from dual-luciferase assays in tobacco leaves. The REN and LUC activities of different combinations of effector (35S::CmMYC2 and 35S::CmJAZ1) and reporter (P*_CmFT527_*, P*_CmFT933_*, P*_CmLeafy2-465_*, P*_CmLeafy2-674_*, P*_CmLeafy3-473_* and P*_CmLeafy3-878_*) constructs were measured; (**e**) Construct details for Y2H assays. *CmJAZ1-3* was cloned into bait plasmid pGBKT7 vectors, while *CmMYC2-2* was inserted into a pGADT7 vector as a prey; (**f**) A yeast two-hybrid assay showing protein interactions. BD is the GAL4 DNA binding domain; AD is the GAL4 activation domain; -TL, SD/-Trp/-Leu; -TLHA, SD/-Trp/-Leu/-Ade/-His; -TLHA + X-α-gal, SD/-Trp/-Leu/-Ade/-His/X-α-gal; Em, empty vector; pGBKT7-53 and pGADT7-T served as positive controls, and pGBKT7-Lam and pGADT7-T served as negative controls.

**Figure 11 ijms-23-06452-f011:**
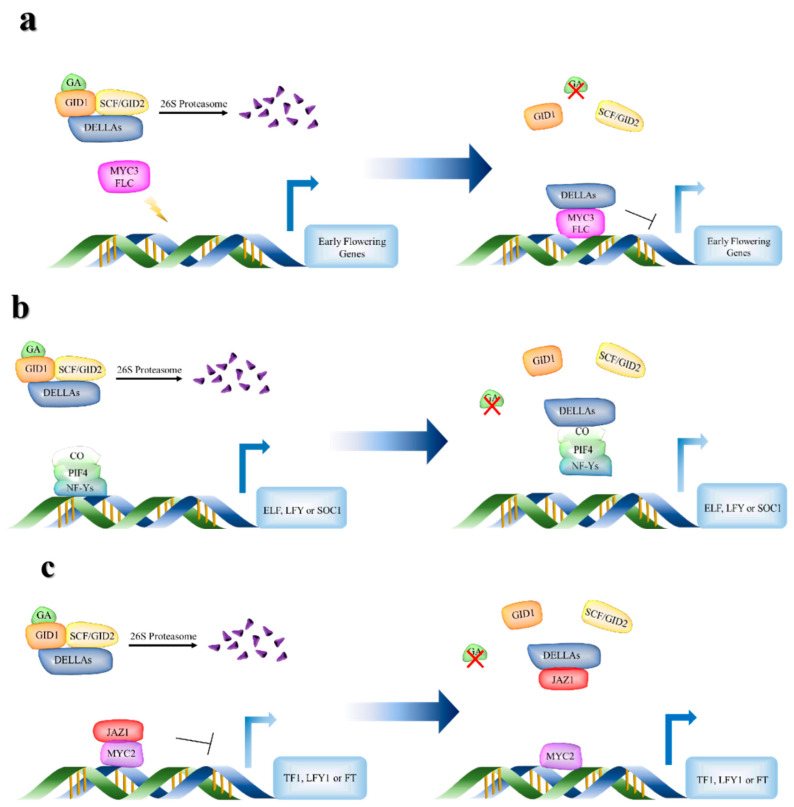
Phytohormone regulated expression model of floral development related genes in two different flower buds of chestnut in pre-winter and post-winter. (**a**) In pre-winter, the higher GA content promoted the increase in GID1 and GID2 expression levels, and inhibited the expression of *DELLAs* gene, further decrease the transcription level of *MYC3* and *FLC*, which was conducive to breaking dormancy and promoting the development of male inflorescence in the two different flower buds; (**b**) In post-winter, the decrease in GAs content led to the decrease in *GID1* and *GID2* expression levels and the increase in *DELLAs* expression levels in IMF, which decrease the expression of *FT*/*LFY*/*SOC1* and other genes, and thus inhibited the differentiation of female flower buds; (**c**) The content of JA-ILE increases significantly and the GAs decreases in CMF in post-winter, which lead to the transcription level of *JAZ1* decreases and the expression level of *DELLAs* increases. The higher DELLAs protein binds to JAZ1 in CMF in post-winter. This competitive binding enables the JAZ1-MYC2 complex to release MYC2, and the higher transcriptional level of *MYC2* initiates the expression of genes related to female flower development, such as *TF1*, *LFY1*, and *FT*.

**Table 1 ijms-23-06452-t001:** Function annotation result statistics.

Database	Number of Annotated Genes
NR	2582
Swissprot	1707
TrEMBL	2461
KOG	2002
KEGG	226
GO	1677

**Table 2 ijms-23-06452-t002:** Screening results of different metabolites.

Group	Significant Difference in the Number of Metabolites	Down Regulation of the Number of Metabolites	Up Regulation of the Number of Metabolites
A2 vs. B2	59	20	39
A1 vs. B1	90	16	74
A2 vs. A1	120	30	90
B2 vs. B1	171	34	137

## Data Availability

Not applicable.

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
