# Peer review of "JAZ1-3 and MYC2-1 Synergistically Regulate the Transformation from Completely Mixed Flower Buds to Female Flower Buds in Castanea mollisima"

_ijms, 2022, doi:10.3390/ijms23126452_

Round 1
Reviewer 1 Report
The manuscript by Cheng et al. “JAZ1-3 and MYC2-1 synergistically regulate the transformation from completely mixed flower buds to female flower buds in Castanea mollisima” is very interesting and experimentally well-demonstrated. They utilized transcriptome sequencing, metabolome analysis and found the key differential genes and differential metabolites involved in regulating the differentiation of female inflorescence primordia of chestnut Authors analyzed the content changes in five kinds of phytohormones (Auxin, Cytokinin, ABA, JA and GA) in the process of female inflorescence primordium differentiation in chestnut mixed flower buds, to reveal the molecular mechanism of regulating the proportion of hermaphrodite inflorescences and the regulation mechanism of endogenous metabolites, especially endogenous hormones. The theme of the work is exciting and emerging. Hence this reviewer feels that this manuscript requires minor revision prior to its publication in International Journal of Molecular Sciences as follows:
Comments
- The authors should cross-check all abbreviations in the manuscript. Initially, define in full name followed by abbreviation.
- The English of manuscript can be polished (minor) and there are few typo errors in the manuscript that can be checked.
- The authors may additionally provide challenges, or prospect of the present study.
- Figure number 2, 4, 6, 8 and 9 quality may be improved (high resolution).
Author Response
Dear reviewer, Thank you for your valuable suggestions. According to your suggestions, we have modified the abbreviation and the corresponding full name in the manuscript; Replace figures 2, 4, 6, 8 and 9 with high-quality pictures again; Finally, the outlook and assumption are put forward in the discussion and summary part. Please refer to the document modification trace.
Best!
Reviewer 2 Report
The manuscript deals with the problem of flower formation in the woody plant C.molissima with an emphasis on identifying factors that play a primary role in the formation of female and male flowers. Since the Chinese chestnut (C. molissima) is of interest as a food crop, the identification and understanding of the reasons that ensure the transition of flower buds to female development is of great practical importance, since it is associated with fruit yield in the chestnut industry. The biological features of the development of flowers in chestnut trees are related to the fact that the laying of flower buds by male and female types are separated in time - before the plants leave for winter dormancy and after winter dormancy. Taking into account the biological features of the formation of flower buds in Chinese chestnut, the authors analyzed the data of transcyptom and metabolomic analyzes of two groups of flower buds of a mixed type of pre-winter and post-winter rudiments of female inflorescences. Based on the results of the analysis, the authors established an important role in the differentiation of flower buds in the post-winter period according to the female type of the ratio between the hormones gibberellins and jasmonic acid, namely, low concentrations of GA and higher JA-ILE. The idea of ​​co-regulation of GA and JA is summarized in the general scheme of differentiation and transition of flower buds to female development. The key role in the regulation of this process is associated with the expression of the JAZ1-3 and MYC2-1 genes. The role of these genes in the regulation of FT gene expression was confirmed by the authors in in vitro experiments using tobacco and yeast transgenesis. In general, the manuscript addresses the important problem of floral differentiation of vegetative buds in woody plants, which undoubtedly represents a big fundamental problem, the solution of which can be directly related to the improvement of the economically valuable characteristics of the Chinese chestnut. The manuscript may be recommended for publication. A small note on this manuscript - the sentence on lines 54-55 is repeated in lines 58-59.

Author Response
Dear reviewer, Thank you very much for your advice and encouragement. We have modified repeated statements and some small errors. Please refer to the document modification trace.
Reviewer 3 Report
The work by Cheng et al. offers an applied utilization of genetic expression techniques to study flower development and the genetic basis of dioecy in Castanea. It successfully merges a natural field variation with lab trials to understand flowers buds development.
A key pro of the study is that this is a well-designed expression analysis that at first glance pushes forward the understanding on how dioecy arises during flower bud deployment. However, a more careful read would invite authors to be more careful at clearly drawing the line at the Introduction (last paragraph, where objectives are stated, L128) and the Discussion sections on what novelty this report conveys given recent review works on dioecy in plants (i.e., please refer to and cite Renner’s synthesis in Nat Plants 2021 7(4):392-402). Furthermore, authors should also discuss more thoroughly reported correlations in dioecious tree species of flower bud phenology with pest resistance (refer to the review Plants 2021 10:2022 and works herein) and abiotic tolerance (refer to the synthesis Front Plant Sci 2020 11:583323 and works herein) traits.
As a second, more technical, point, methods, results and data interpretation are generally coherent, and conclusions are undeniably well supported. Still, what are the chances of flower bud development to exhibit a more polygenetic basis beyond the DEGs (with more subtle effects), not to mention a stronger environmental interaction (i.e., environmentally conditional-dependent DEGs). Authors must refer from the very begging to these possibilities, and not just rely on expression differences in terms of the flowering development found at the studied treatments. A general recommendation is to first comment on these alternative drivers, or at least envisioning is consideration as part of this or future studies. Furthermore, please prospect, as perspectives at the end of the Discussion section (L687), the utility of testing the current repertoire of DEGs novel environments (see Front Environ Sci 2021 9:619092). I encourage authors to explicitly embrace these points and discuss the statistical power explicitly in the Discussion section by adding a paragraph on possible caveats, including the standing diversity within the analyses sample set for the DEGs (refer to Genes 2021 12:783).
Finally, also as discussion point, please comment whether the current DEGs may serve as future targets for gene editing strategies trying to induce precocity in otherwise perennial tree plantations, or boost reproductive sterility in genetically engineered plantation forests (as for some Castanea species). Mention these ad hoc hypotheses explicitly at the end of the discussion section. Specifically, how may the current study be couple with genomic prediction and speed breeding strategies targeting woody perennials (as already envisioned for tree species by Dario Grattapaglia in the lower right corner of figure 1 in the review Front Plant Sci 2018 9:1693, as well as by Amy Brunner in the short review at Front Plant Sci 2018 9:1671).
Author Response
Dear reviewer, Thank you for your in-depth analysis and suggestions. In particular, the document NAT plants 20217, (4): 392-402 has given me a lot of inspiration and help. In the preface and discussion, we have added information about possible mining of DEGs and metabolome joint analysis and existing problems; In the discussion part, it is expected that GWAS and GP will be combined with gene editing technology in the future to edit the related genes of DEGs traits, so as to improve the yield of woody plants, shorten the infancy of fruit trees, change the sterility of trees, and achieve the goal of tree breeding. Best!
Round 2
Reviewer 3 Report
Authors have carefully argued and addressed my recommendations. From my side, the work is suitable given the scope of IJMS.